**Triggering effects of large topography and boundary layer turbulence on convection over the Tibetan Plateau**

**Xiangde Xu[1], Yi Tang[1,2], Yinjun Wang[1], Hongshen Zhang[3], Ruixia Liu[4] and Mingyu Zhou[5]**

[1] State Key Laboratory of Severe Weather, Chinese Academy of Meteorological Sciences, Beijing, China.

[2] School of Environmental Studies, China University of Geosciences, Wuhan, China.

[3] Peking University, Beijing, China

[4] CMA Earth System Modeling and Prediction Centre (CEMC), Beijing, China

[5] National Marine Environmental Forecasting Center, Beijing, China

Corresponding author: Yinjun Wang (pbl_wyj@sina.cn) and Hongshen Zhang (hsdq@pku.edu.cn)

## Abstract

In this study, we analyze the diurnal variations and formation mechanism of low clouds at different elevations. We further discuss whether there exists triggering mechanism for convection over the Tibetan Plateau (TP), and whether there is an association among low air density, strong turbulence and ubiquitous "popcorn-like" cumulus clouds. The buoyancy term (BT) and shear term (ST) over the TP are significantly greater than those at the low elevation, which is favorable for the formation of increasing planetary boundary layer height (PBLH), and also plays a key role in the convective activities in the lower troposphere. From the viewpoint of global effects, the triggering of convection by boundary layer dynamics is analyzed over TP but also in the Northern Hemisphere over the Rocky Mountains. It is found that ST and BT are strong over both high elevation regions. The strong thermal turbulence and large scale ascending motion jointly results in obvious positive value of PBLH-LCL under low relative humidity (RH) condition over the TP. The obvious large-scale subsidence on both sides of the Rocky Mountain especially the western side leads to inversion above PBL and lower RH within the PBL, which further lead to negative value of PBLH-LCL and decreased low cloud cover (LCC) in most part of Rocky Mountain. The slightly greater than zero PBLH-LCL corresponds spatially to increased LCC in the partial regions of central Rocky Mountain. Thus less LCC is generated at the Rocky Mountains compared to the TP.

## Introduction

The Tibetan Plateau (TP), which resembles a "third pole" and a "world water tower", plays an important and special role in the global climate and energy–water cycle (Xu et al., 2008; Wu et al., 2015). The TP covers a quarter of China. Additionally, the average altitude of the TP is 4000 meters, reaching 1/3 of the tropopause height, so it is called the "World Roof". Cumulus convection over the TP transfers heat, moisture and momentum into the free troposphere, which can impact the atmospheric circulation regionally and globally (Li and Zhang, 2016; Xu et al., 2014) and reveals the important "window effect" for the transfer and exchange of global energy and water vapor over the TP. It is a dynamic effect caused by the special heat source that constitutes the "window effect" and "thermally driven" mechanism over the TP.

The results of the second Tibetan Plateau Experiments (TIPEX II), which were carried out in 1998, show that the strong convective plumes within PBL observed by sodar and a frequently occurred deep mixed layer (>2 km) can lead to ubiquitous "popcorn-like" cumulus clouds in Dangxiong, as proposed by Zhou et al. (2000), and Xu et al. (2002) came up with a comprehensive physical pattern of land-air dynamic and thermal structure on the TP (Xu et al., 2002; Zhou, 2000). The previous studies have done many valuable researches on the triggering mechanism of moist convection over moist and dry surfaces based on atmospheric observations and simulations (Ek and Mahrt, 1994; Findell and Eltahir, 2003; Gentine et al., 2013). For dry surface, the weak stratification and strong sensible heat flux result in the rapid growth of PBLH so that the relative humidity at the top of the boundary layer $RH_{top}$ increases rapidly, which favors the formation of clouds. For moist surface, strong stratification and evaporation (small bowen ratio) not only cause slow growth of PBLH but also increase the mixed layer specific humidity and $RH_{top}$, which favor the formation and development of clouds. Taylor et al. (2012) found that the afternoon rain falls preferentially over soils that are relatively dry compared to the surrounding area, especially for semi-arid regions. Guillod et al. (2015) reconciled spatial and temporal soil moisture effects on the afternoon rainfall. They showed that afternoon precipitation events tend to occur during wet and heterogeneous soil moisture conditions, while being located over comparatively drier patches. Tuttle et al. (2016) showed the empirical evidence of contrasting soil moisture–precipitation feedbacks across the United States, and they found that soil moisture anomalies significantly influence rainfall probabilities over 38% of the area with a median factor of 13%. Findell et al. (2003) analyzed the model results over dry and wet soils in Illinois. They summarized the predictive capability of rain and shallow clouds by using the convective triggering potential (CTP) and a low-level humidity index, with $HI_{low}$ as measures of the early morning atmospheric setting. Our previous studies pointed out that the developments of these cumulus clouds are related to the special large scale dynamic structure and turbulence within PBL over the TP (Xu et al., 2014; Wang et al., 2020). In addition, Wang et al., (2020) pointed out that, despite the same relative humidity between eastern China and the TP, the lower temperature over the TP results

in a lower lifting condensation level. With the same surface sensible heat flux, lower air density over the TP results in a larger buoyancy flux and a deeper boundary layer. All the above results indicate the topography of the TP plays a major role in increasing the occurrence frequency of strong convective clouds (Luo et al., 2011). This conclusion is consistent with the viewpoint of Flohn (1967) who emphasized the chimney effect of the huge cumulonimbus clouds on heat transfer in the upper troposphere.

The TP is one of the regions in China that is featured with high frequency of cumulus clouds, and the development of a cumulus system is related to both the turbulence and special dynamical structure in the PBL over the TP. The vertical motion over the TP is associated with anomalous convective activities. However, as Li and Zhang (2016) mentioned, the details of PBL processes are not very clear. The same is true for the diurnal variations and formation mechanism of low clouds over the TP and low elevation regions. The different variation characteristics of these low clouds at different elevations and regions also need to be discussed and analyzed. Moreover, we need to investigate whether there exists "high efficiency" triggering mechanisms for convection over the TP, and whether there is an association among low air density, strong turbulence and ubiquitous "popcorn-like" cumulus clouds. Is there also strong turbulence at higher elevation regions with lower air density in the globe? What is the impact of the large scale vertical motions on clouds? Because both the TP and Rocky Mountains are high elevation regions covering large mid-latitude areas, we select these two typical regions to make a deep analysis. Unlike our previous paper by Wang et al. (2020), in this study we mainly focus on the comparison between these two regions to analyze the above scientific questions.

**2 Observational and reanalysis data**

We use in situ measurements of temperature (T) and relative humidity (RH) at 2 m height, surface pressure data every hour, and low cloud cover (LCC) every three hours from 2402 automatic weather stations from June to August of 2010-2019 in China. LCC here refers to the fraction of the sky covered by low clouds as estimated by human observers, including five cloud types: nimbostratus (Ns), stratocumulus (Sc), stratus (St), cumulus (Cu), and deep convection (DC). These surface observation datasets are provided by China National Meteorological Information Center.

In addition, we use the hourly 0.25° x 0.25° ERA5 reanalysis surface-layer data in summer (June 1 to August 31) from 2010 to 2019 (Hersbach et al., 2020).

We use more than 4 years (from June 15 2006 to August 31 2010) of the satellite (CloudSat radar and Calipso lidar)-merged cloud classification product 2B-CLDCLASS-lidar to calculate the mean LCC with 1°x1° resolution at about 2:00 pm and 2:00 am LT in summer. The introduction of this product and details of the LCC calculation methods are summarized in Sassen and Wang (2008) and Wang et al (2020).

We use a Gaofen 4 (GF 4) visible satellite image with a spatial resolution of 50 m on August 4 of 2020 to show the organized structures (cellular convection) in southeastern TP, as shown in Figure 1. GF 4 is a geostationary earth observation satellite in the Gaofen series of Chinese civilian remote sensing satellites. We also use

the 1 year (from June 1 to August 31 of 2016) geostationary satellite Himawari-8
retrieval product (cloud top height) over land in East Asia.
In this study, we also use temperature (T) at 2 m height, relative humidity (RH) at
2 m height, surface pressure and planetary boundary layer height (PBLH) from ERA5
reanalysis data from 2010 to 2019. To be specific, the above four variables represent
hourly averaged values for each month (24 values in total for a month). The lifting
condensation level (LCL) is calculated by the method proposed by (Romps, 2017).
Using sensible heat flux $H$, Northward turbulent surface stress $\tau_y$ and Eastward
turbulent surface stress $\tau_x$ from ERA5 reanalysis data, we calculate the buoyancy term
(BT) ($g/\theta_v \overline{w'\theta_v'}$) and shear term (ST) ($-\partial\overline{u}/\partial z\,\overline{u'w'}$) in the TKE equation for each grid.
Both of these two terms can be used to analyze the effect of boundary layer
turbulence in surface layer on convection. The details of the method for computing
BT and ST are as follows:
The shear term (ST) ($-\partial\overline{u}/\partial z\,\overline{u'w'}$) and buoyancy term (BT) ($g/\theta_v \overline{w'\theta_v'}$) in the
TKE equation maintain the turbulent motions. In order to simplify calculations, the
x-axis is directed along the average wind. Assuming horizontal homogeneity and no
mean divergence, the TKE equation is written as
$$\frac{\partial \overline{e}}{\partial t} = \frac{g}{\theta_v}\overline{w'\theta_v'} - \overline{u'w'}\frac{\partial \overline{u}}{\partial z} - \frac{\partial\left(\overline{w'e}\right)}{\partial z} - \frac{1}{\rho}\frac{\partial\left(\overline{w'p'}\right)}{\partial z} - \varepsilon. \tag{1}$$

The left side of eq. (1) is the local time variation $\partial\overline{e}/\partial t$, and the terms on the
right-hand side of eq. (1) describe the buoyancy and shear energy production or
consumption, turbulent transport of $\overline{e}$, pressure correlation and viscous dissipation
(Stull, 1988).
Here we use eq. (2) to calculate the virtual potential temperature $\theta_v$, and $\overline{w'\theta_v'}$ is
derived from eq. (3). Finally, we derive BT.
$$\theta_v = T\left(1 + 0.608q\right)\left(\frac{p_0}{p}\right)^{\frac{R}{c_p}}, \tag{2}$$

$$H = \rho c_p \overline{w'\theta_v'}, \tag{3}$$

Where $g$ = 9.8 m s$^{-2}$ is the gravitational constant, and $H$ (W m$^{-2}$) is the sensible
heat flux, $\rho$ (kg m$^{-3}$) is the air density, R is the specific gas constant for dry air, $c_p$
(=1004 J kg$^{-1}$ K$^{-1}$) is the specific heat of air at constant pressure, $T$ is the air
temperature at 2 m height, $q$ is the specific humidity at 2 m height, $p_0$ and $p$ are
standard atmospheric pressure and surface pressure, respectively.
The wind shear is determined from heat flux $H$ and momentum flux $\tau$ obtained
from the ERA5 reanalysis data. Because we cannot directly obtain the $\tau$ from ERA5
product list, we need to use eq. (4) to calculate $\tau$.
$$\tau = \sqrt{\tau_x^2 + \tau_y^2}, \tag{4}$$

According to Monin Obukhov similarity theory wind shear is given as

$$\frac{\partial \overline{u}}{\partial z} = \phi_m(\zeta)\frac{u_*}{\kappa z},$$
(5)

Where $\phi_m$ is the Monin Obukhov stability function for momentum, $u_*^2 = \tau/\rho$.
The von Karman constant $\kappa = 0.4$, $\overline{u}$ is the horizontal wind speed in the surface
layer.

$\zeta = z/L$ with $z$ = height and $L$ = Obukhov stability length defined as in Gryanik et
al. (2020) as
$$\zeta = \frac{z}{L}, L = -\frac{(\tau/\rho)^{3/2}}{\kappa (g/\theta_v)(H/\rho c_p)}.$$
(6)

$\phi_m$ is the Monin Obukhov stability function, here we use eq. (7) and eq. (8) for
stable and unstable conditions to derive $\phi_m$ (Dyer, 1974),

$\phi_m = 1 + 5\zeta, (\zeta > 0)$                    (7)

$\phi_m = (1 - 16\zeta)^{-1/4}, (\zeta < 0)$      (8)

Then we use eq. (9) to derive $-\overline{u'w'}$. Finally, we derive ST.
$\overline{u'w'} = -\tau/\rho.$                         (9)


## 3 Results

Figure 2 shows the spatial distribution of over-land low cloud cover (LCC) in
China from June to August of 1951-2019. Compared to the low LCC in eastern China,
the high value areas of LCC are mainly located in the mid-eastern TP and the area of
the upper Yangtze River Valley. But low LCC is also identified in western and
northern parts of TP. We will make a further discuss about it in subsequent paragraphs.
Using four years of CloudSat-Calipso satellite data, Li and Zhang (2016) confirmed
that the climatological occurrence of cumulus over the TP is significantly greater than
that in mid-eastern China on the same latitude. The elevated land surface with strong
radiative heating makes the massive TP a favorable region for initiating convective
cells with a high frequency of cumulonimbus and mesoscale convective systems
(Sugimoto and Ueno, 2012). As a strong heat source, the TP has frequent convective
activities in summer. During the TIPEX Ⅱ in 1998, the long and narrow thermal
plume corresponding with vigorous cellular convection on micro-scale was observed
by sodar in Dangxiong. As shown in Figure 1, the shallow convective clouds on a
horizontal scale from hundreds of meters to several kilometres over the southeastern
TP (92.7-96.2E, 29.5-31.3N) are probably related to the organized eddies on the
meso-scale and micro-scale over the TP. The cloud fraction over the southeastern TP
is about 31.3%.

As shown in Figure 3, in general, LCC increases with increasing elevation. The
median of $LCC_H$ is significantly greater than those of $LCC_L$ and $LCC_M$ throughout the
day. The diurnal variations of $LCC_L$ and $LCC_M$ are generally distributed in unimodal
pattern, with the maximum appearing at 2:00 pm Beijing time (median $LCC_L$ = 37%,

LCC$_M$ = 38%) and low values (~20%) are maintained during the night. The diurnal variation of LCC$_H$ presents a bimodal curve with the maximum appearing at 5:00 pm Beijing time (median LCC$_H$ = 69%) and the secondary local maximum appearing at 8:00 am Beijing time (median LCC$_H$ = 61%). Compared to the low elevation, the interquartile ranges (IQRs) of LCC$_H$ are smaller than those of LCC$_L$ and LCC$_M$, which imply the LCC$_H$ maintains high values during the day. To further confirm and compare the above results with in situ measurements, using ERA5 LCC data, we also add Figure S1 to show the diurnal cycle of LCC in summer in East Asia and North America in the supplementary material.

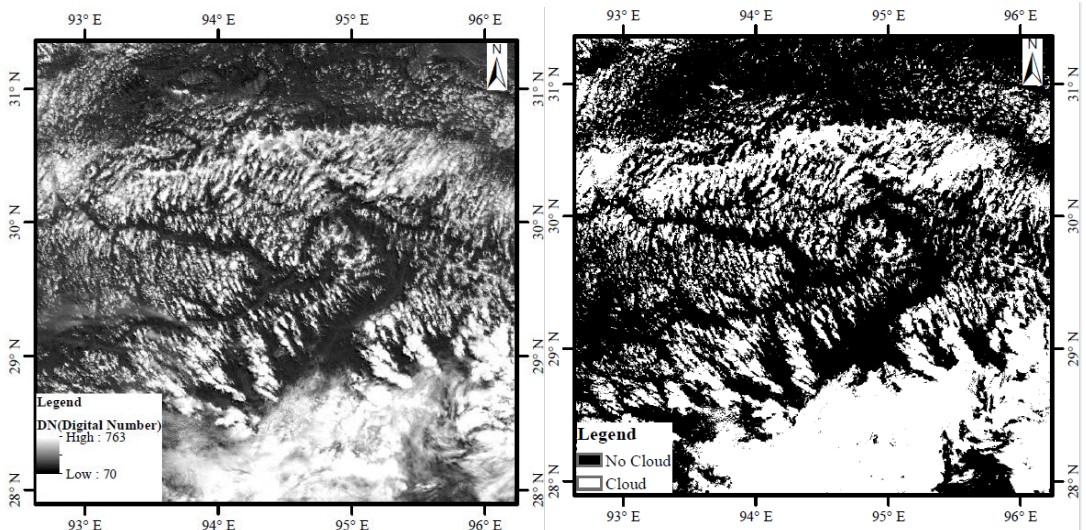

Figure 1. The (a) digital number (DN) and (b) spatial distribution of cloud in southeastern TP from geostationary earth observation satellite Gaofen 4 (GF4) at 12:00 pm Beijing time (about 10:20 am local time) on August 4 of 2020. Here we simply use DN = 250 as a threshold. All the grids in Figure (a) are divided into two classes (DN > 250, cloud; DN < 250, no cloud), and then we give Figure (b).

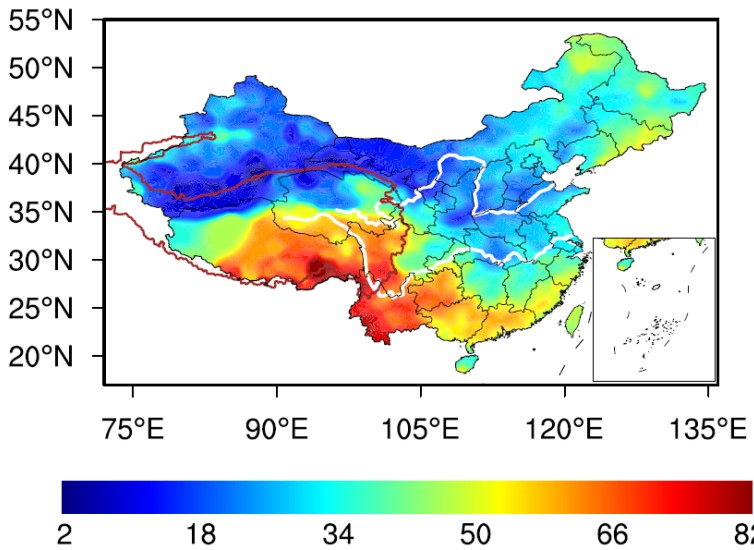

Figure 2. The summer mean LCC derived from surface observations from 1951 to 2019 in China. The thick

red contour denotes the 2.5 km topography height referred to as the TP. The white lines located in northern and southern parts of China denote the Yellow and Yangtze River, respectively.

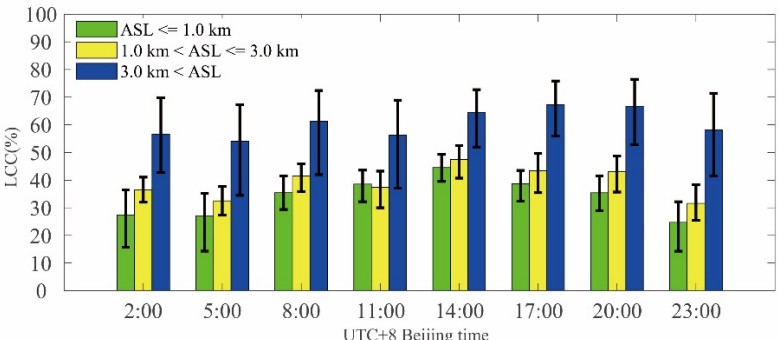

Figure 3. The diurnal cycle of LCC in summer from 2010 to 2019 at different altitudes above sea level (ASL): ASL ≤ 1.0 km (LCC$_L$), 1.0 km < ASL ≤ 3.0 km (LCC$_M$), and 3.0 km < ASL (LCC$_H$). It should be noted that all the sites are ranged from 27N to 40N in China, and each sample is derived from monthly mean LCC at a particular time in summer for each site. The bar and error bar represent the median values and interquartile ranges (IQRs) of LCC, respectively. The subscripts L, M and H of LCC denote the low, mediun and high clouds, respectively.

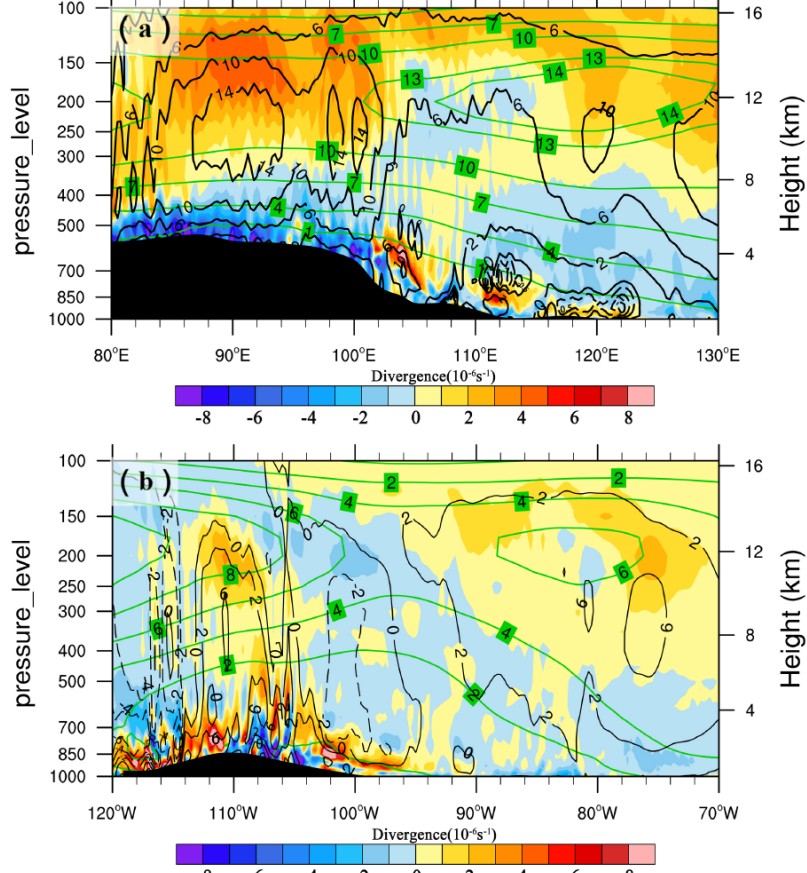

229

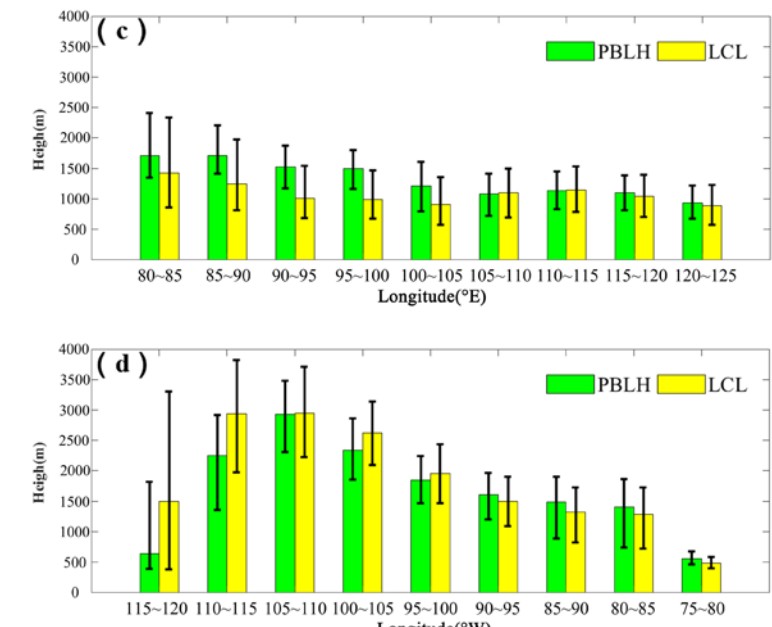

230

Figure 4. Vertical distribution of summer mean divergence ($10^{-6}$ s$^{-1}$) (shaded) at 2:00 pm local time from 2010 to 2019 at the latitude across sections from 30N to 35N in (a) East Asia and (b) North America. The green and black contours denote the summer mean U- (m s$^{-1}$) and W- ($10^{-2}$ m s$^{-1}$) wind components with the zonal circulations, respectively. The solid and dashed contour lines represent the positive and negative values, respectively. The black shaded areas represent topography. Figure (c) and (d) are the PBLH (green) and LCL (yellow) versus longitude in East Asia and North America, respectively. The bar and error bar represent the median values and interquartile ranges (IQRs), respectively.

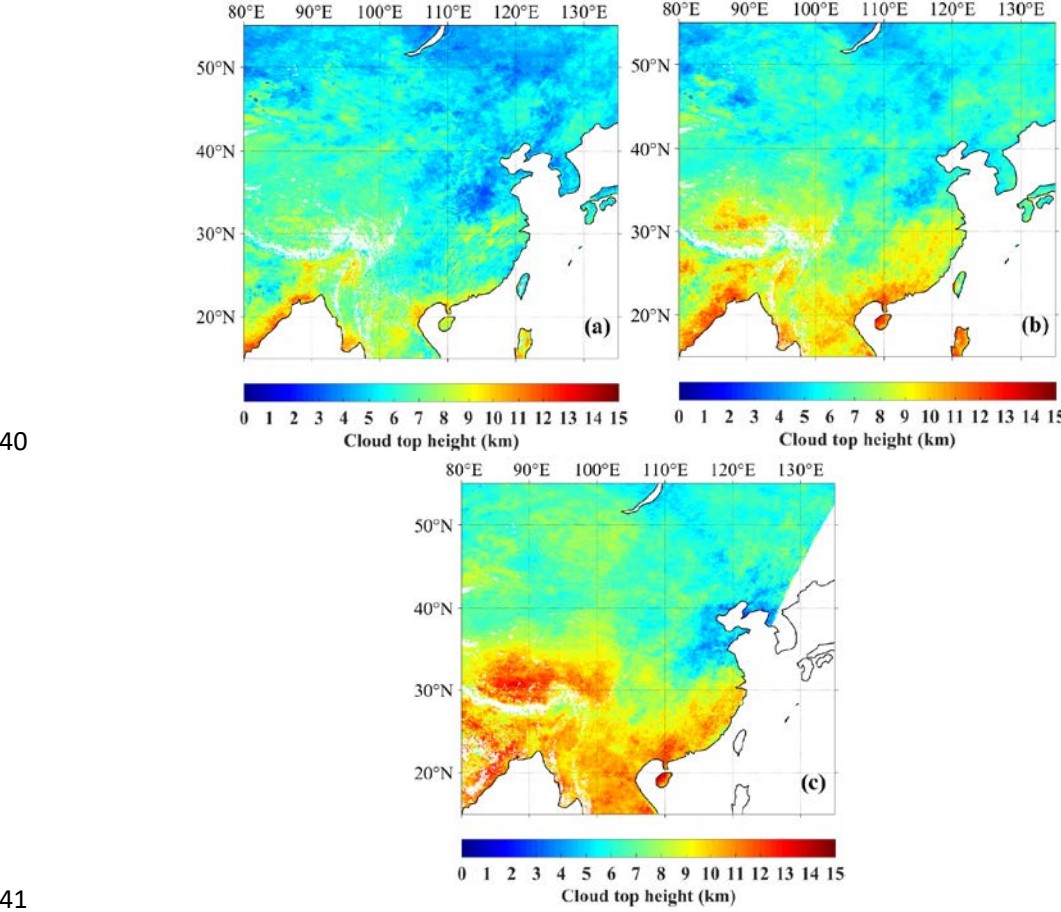

Figure 5. The median cloud top height derived from the Himawari-8 retrieval product at three Beijing times: (a) 2:30 pm±0.5h (b) 4:30 pm±0.5h (c) 6:30 pm±0.5h from June to August in 2016 over land in East Asia. Missing data are shaded in white color.

On the other hand, we note that, there is no obvious trend of decreasing LCC over the TP from late afternoon to evening as shown in Figure 3. Based on the spatial distribution of topography in the Northern Hemisphere as shown in Figure 7 (a), it is clear that both the TP (27-40N, 70-105E) and Rocky Mountains (27-40N, 103-120W) in North America are two large areas with high elevations in mid-latitude regions in the Northern Hemisphere, so here we select these two typical large topography regions to analyze the triggering effects of large topography and related dynamical structure within the boundary layer on convective clouds. As shown in Figure 4 (a), in general, there are obvious large scale ascending motions from middle troposphere (~500 hPa) to upper troposphere (~200 hPa) over the TP. The convergence in the middle troposphere (the blue shaded areas) and the divergence in the upper troposphere (the orange shaded areas) are usually associated with the deep convection over the TP. Figure 4 (c) shows there are generally positive PBLH-LCL (~500 m) over the TP, and the median and IQR of PBLH are close to those of LCL in East China. These results are consistent with the conclusions proposed by Xu et al. (2014) and Wang et al. (2020). In contrast, Figure 4 (b) shows there are only weak large scale ascending motions from near surface layer to the middle troposphere over the Rocky Mountains. The large-scale subsidence on both sides of the Rocky Mountains

especially the western side can lead to inversion above PBL and lower RH within the PBL, which can be verified by the vertical distribution of the $d\theta_v/dz$ and RH at the latitude across sections from 30N to 35N over the Rocky Mountains in Figure S2. There exists a high value center of $d\theta_v/dz$ at about 950 hPa (or 850 hPa) on western (or eastern) side of the Rocky Mountains, and the RH within the PBL is generally less than 55%. The former restricts the growth of PBLH during the day, while the latter leads to an increased LCL. Thus, negative PBLH-LCL is identified on both sides of the Rocky Mountains (30-35N, 110-120W and 30-35N, 100-105W), especially for the western Rocky Mountain (30-35N, 110-120W) with strong large-scale subsidence, as shown in Figure 4 (d).Dynamic processes of vapor transport are generated because of the thermal structure of the TP, which is similar to the conditional instability of the second kind (CISK) mechanism of tropical cyclones (Smith, 1997). It should be pointed out that there are large scale descending motions at 500 hPa in part of the western TP and Qaidam Basin as shown in Figure S3, which lead to less LCC in these regions compared to the other parts of the TP, as shown in Figure 2. In addition, the meteorological stations in the northern TP (34-36N, 80-90E) are scarcely and unevenly distributed, and therefore the low LCC in the Taklamakan Desert leads to false low LCC values in the northern TP (80-90E, 34-36N), as shown in Figure 2. In fact, there are high LCC in these regions as shown in Figure 7 (e).Figure 5 shows the spatial distribution of day time variations of cloud top height in summer. Compared to eastern China at the same latitude, the cloud top height increases significantly from 2:30 pm (~7 km) to 6:30 pm (~14 km) over the TP. The cloud top height approaches the tropopause (~14 km) in the evening, which implies the frequent occurrence of deep convective clouds at this time. This result is consistent with the observation of millimeter-wave radar in Naqu (Yi and Guo, 2016).

By comprehensively analyzing the TIPEX II sodar data, Xu et al. (2002) and Zhou et al. (2000) found that, with narrow upward motion and time scale from 1.2 h to 1.5 h, the maximum upward motion of the thermal turbulence was identified at the height of about 120 m above the surface, with the vertical speed up to 1 m s$^{-1}$. They also found symmetrical and wide downward motion areas on either side of the narrow upward motion zone. The question arises as to whether there is a relationship between the formation and evolution of frequent "pop-corn-like" convective clouds and micro-scale thermal turbulence in the atmospheric convective boundary layer over the TP. Xu et al., (2012) speculate these low clouds are probably initiated by strong thermal turbulence under low air density conditions. Compared to the low elevation in eastern China, the increased thermal turbulence associated with low air density over the TP leads to the different turbulence characteristics of the convective boundary layer (CBL). The CBL is mainly driven by buoyancy heat flux, and the thermal turbulence with organized thermal plume is not totally random (Young, 1988a; Young, 1988b). The strong BT and ST over the TP play key roles in the convective activities in lower troposphere.

By using the statistical results from sodar data in the TIPEX II, Zhou et al. (2000) calculated the BT and ST at the height of 50 m under strong convection conditions in Dangxiong (located at central TP). The results indicate that the BT is comparable to

ST. Both the thermodynamic and dynamic processes have important influences on the convective activities. Both the BT and ST in the surface layer in Dangxiong are almost an order of magnitude greater than those at low elevation given by Brummer (1985) over North Sea and Weckwerthet et al. (1997) in Florida. Direct measurements from the Third Tibetan Plateau Experiments (TIPEX III) also confirmed that surface buoyancy flux over the TP is significantly larger than that in eastern China (Zhou, 2000; Wang et al., 2016). Both the sodar data in TIPEX II and boundary layer tower data in TIPEX III showed contributions of BT and ST to the turbulent kinetic energy in the lower troposphere are larger over the TP than over the southeastern margin of the TP and the low-altitude Chengdu Plain (Zhou, 2000; Wang et al., 2015). Thus one might ask the question what is the relationship between high frequent low cloud and the above physical quantities (e.g. turbulence structure, temperature and humidity) under low air density conditions over the TP? The physical mechanism should be discussed and analyzed. In addition, at low elevation in eastern China, the question arises as to whether or not the variations of PBLH and LCL favor the formation and development of low clouds.

As shown in Figure 6 (a), compared to the low elevation, there is larger LCC (LCC > 50%) over the TP (ASL > 3 km) under low $RH_{2m}$ condition ($RH_{2m} < 40\%$). In contrast, larger LCC mostly corresponds to higher $RH_{2m}$ condition at low elevation, which is consistent with our common sense. The above interesting phenomenon can be explained by the differences of PBLH-LCL between the TP and low elevation regions on summer afternoons. These differences are mainly attributed to following two mechanisms. The first mechanism is that, with a similar sensible heat flux, the lower air density over the TP leads to greater surface buoyancy flux (or BT) as shown in Figure 6 (c), which is conducive to the increase of PBLH over the TP. Figure 6 (d) shows great ST over the TP, which is mainly attributed to large wind speed. Although here we only show the ST in the surface layer, strong wind shear in the boundary layer probably also plays a role in increasing PBLH over the TP. On the other hand, the second mechanism is that, with a similar RH, Wang et al. (2020) have indicated that, compared to the low elevation in eastern China, the lower temperature over the TP leads to a lower LCL. Together these two mechanisms lead to a greater (PBLH-LCL) difference over the TP on summer afternoons, which increases the probability of air parcels reaching the LCL and forming clouds as shown in Figure 6 (b). In most cases, the positive value of PBLH-LCL as well as the great BT and ST over the TP corresponds with larger LCC (LCC > 50%) under low $RH_{2m}$ condition ($RH_{2m} < 60\%$), which implies the enhanced local LCC is relevant to the diurnal variation of the PBL process. In contrast, for the eastern China, in most cases, the increased LCC (LCC > 50%) generally corresponds with high $RH_{2m}$ ($RH_{2m} > 60\%$), and the LCC is not significantly correlated with PBLH-LCL or BT and ST, which implies the other factors besides the PBL process (e.g. large scale ascending motion) play a more important in LCC.

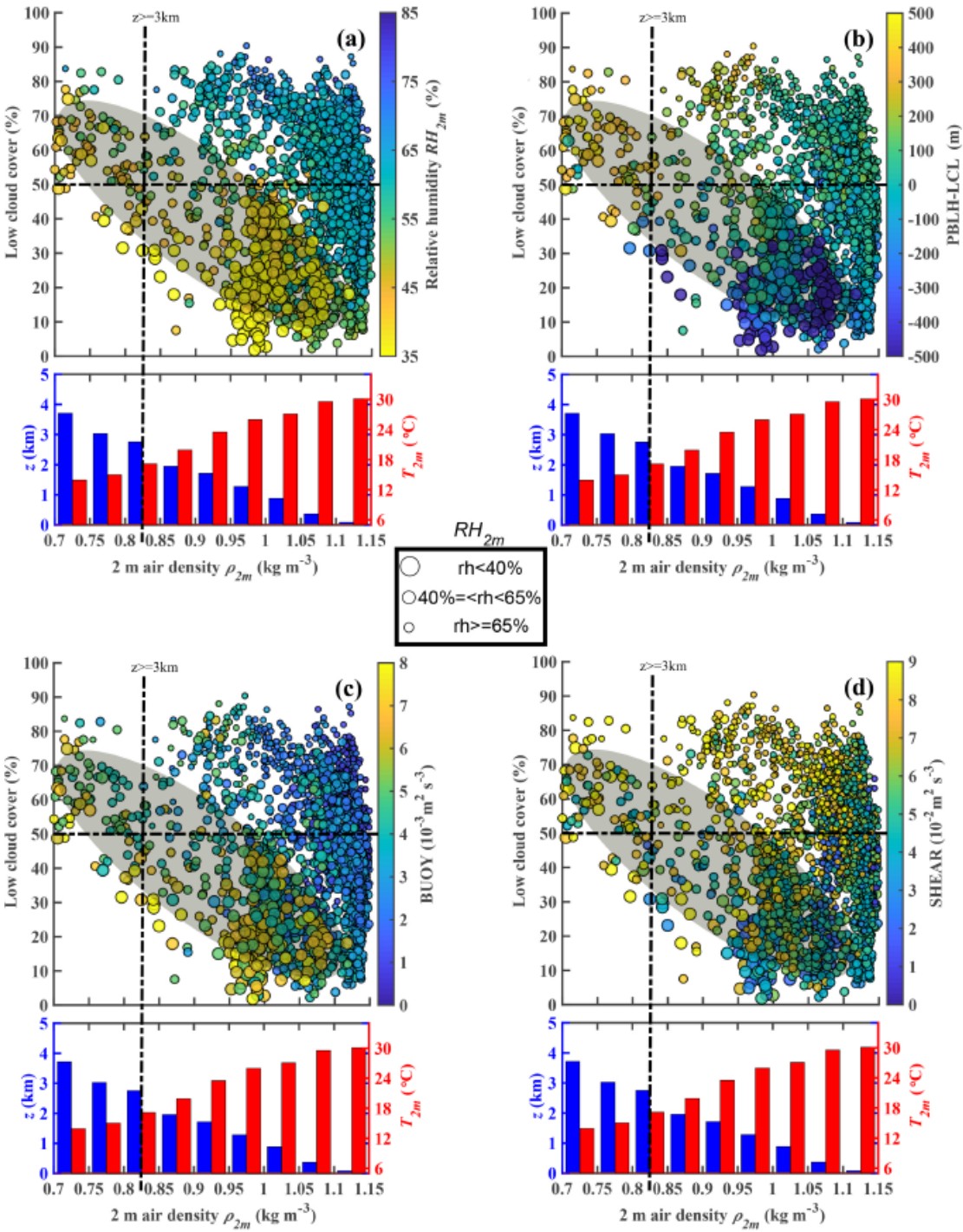

Figure 6. The relationships among monthly means of low cloud cover LCC, $\rho_{2m}$ and (a) $RH_{2m}$, (b) PBLH-LCL, (c) BT and (d) ST at 2:00 pm Beijing time from 2010 to 2019 in summer in China. The samples are divided into three groups: $RH_{2m} >= 60\%$ (small size dots), $60\% > RH_{2m} >= 40\%$ (median size dots) and $RH_{2m} < 40\%$ (large size dots). The LCC, $T_{2m}$ and $RH_{2m}$ are observed by in situ measurements, and PBLH, LCL, BT and ST are derived from ERA5 reanalysis data. Here we use the nearest neighbor gridding method to derive PBLH, LCL, BT and ST at each site. The blue and red histograms show the surface elevation $z$ (blue) and air temperature at 2 m ($T_{2m}$)

(red) as functions of 2 m air density ($\rho_{2m}$). The dots with lower $RH_{2m}$ ($RH_{2m} < 40\%$)
are mostly distributed within grey shaded elliptic regions as shown in Figure 6 (a)-(d).
Figure 7 (d) shows the mean spatial distribution of PBLH – LCL in the Northern
Hemisphere from June to August of 2010-2019. The TP (27-40N, 70-105E) and
Rocky Mountains (27-40N, 103-120W) are two typical large topography regions in
the Northern Hemisphere, and the mean PBLH – LCL over the TP and Rocky
Mountains are 376.7 m and -101.9 m, respectively.
Figure 7 (b)-(c) shows the spatial distribution of ST and BT in the Northern
Hemisphere from June to August of 2010-2019, respectively. The effect of strong
thermal turbulence results in obvious positive values of PBLH – LCL at high
elevation regions under low air density conditions in the Northern Hemisphere (BT =
0.008 $m^2$ $s^{-3}$, PBLH – LCL = 376.7 m over the TP and BT = 0.011 $m^2$ $s^{-3}$, PBLH –
LCL = -101.9 m over the Rocky Mountains). Figure 7 (b) also shows that there are
strong STs at these two high elevation regions (ST = 0.087 $m^2$ $s^{-3}$ over the TP and ST
= 0.085 $m^2$ $s^{-3}$ over the Rocky Mountains). Both the BT and ST increase significantly
at high elevation due to low air density compared to those at low elevation. The above
results enlighten us on thinking about whether the triggering effects of large
topography and boundary layer turbulence, which reflect the special surface
characteristics in the boundary layer at high elevation regions under low air density
conditions, can be applicable for any large topography in the globe, including TP and
other regions (e.g. Rocky Mountains).
Figure 8 shows a conceptual model of the atmosphere from the near-surface to
upper troposphere over the TP. Compared to the low elevation, the TP is characterized
by higher PBLH and lower LCL because of strong BT and ST, which is favorable for
the formation of shallow clouds in the afternoon. Meanwhile, the large scale
ascending motion over the TP results in the transition from shallow clouds to deep
convective clouds in the late afternoon and evening.

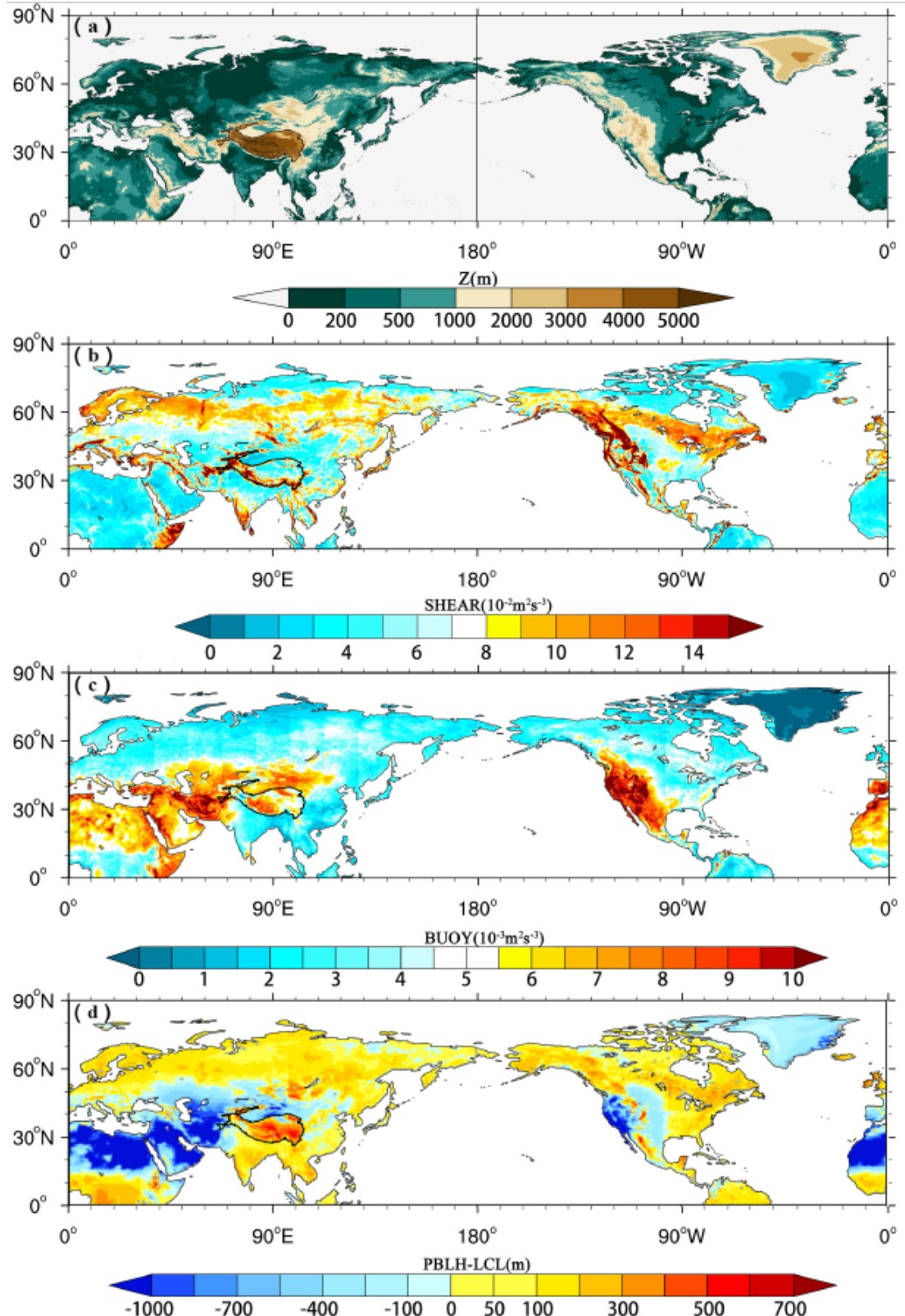


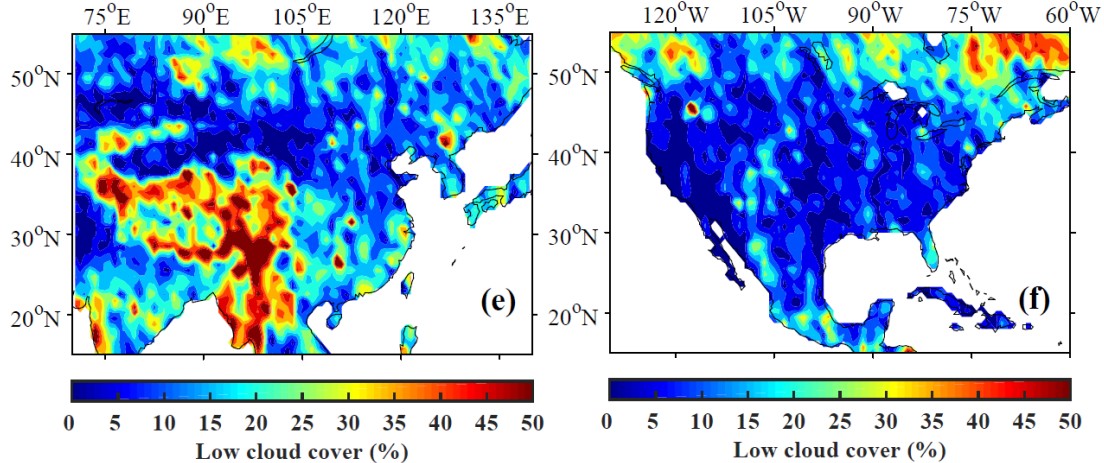

Figure 7. The spatial distribution of (a) ground level elevation (m), (b) ST ($10^{-2}$ m$^2$ s$^{-3}$), (c) BT ($10^{-3}$ m$^2$ s$^{-3}$), and (d) PBLH-LCL (m) derived from ERA5 reanalysis data at local time 2:00 pm in the Northern Hemisphere in summer. Figure (e) and (f) are the summer mean LCC (%) derived from cloudsat satellite data at local time 2:00 pm in eastern Asia and North America, respectively.

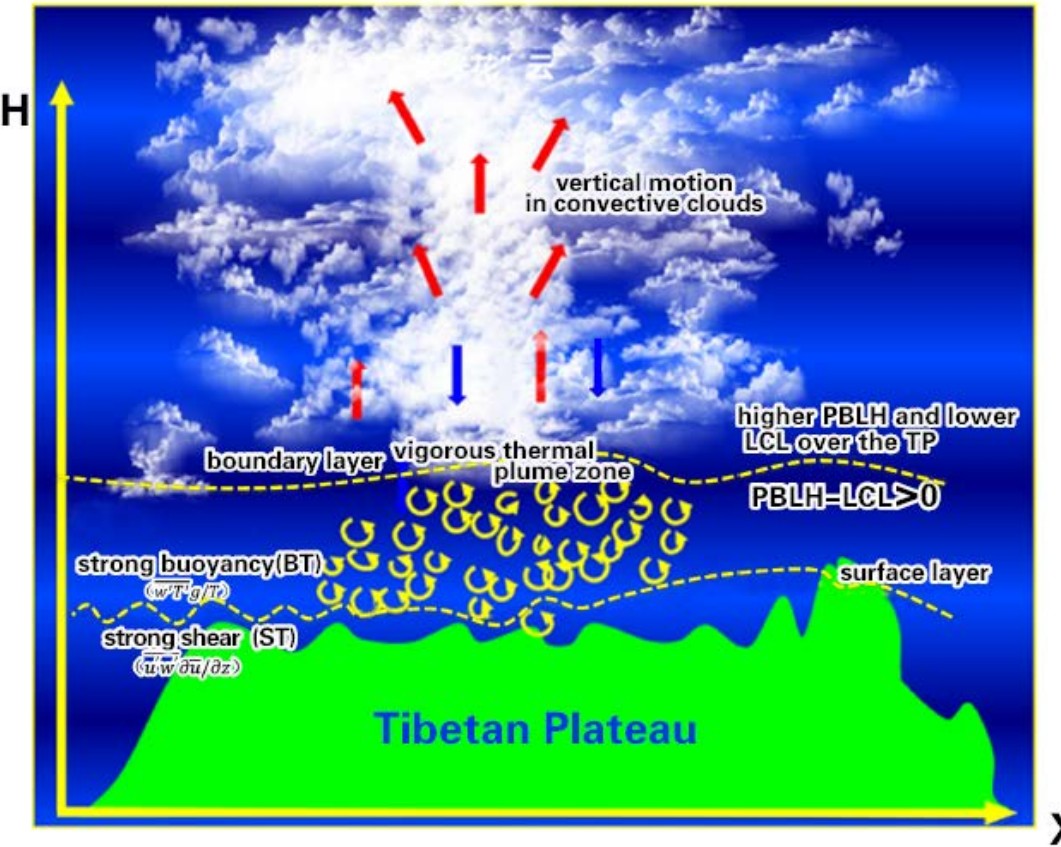

Figure 8. The characteristics model of boundary layer turbulence related to "high efficiency" triggering mechanisms for convection over the TP.

## 4 Conclusions and further discussion

In this study, we focus on the triggering effects of large topography and boundary layer turbulence over the Tibetan Plateau on convection. The topography of the TP plays a major role in the increased occurrences of convective clouds. Our results further confirm the conclusions from Wang et al. (2020), which found that the difference PBLH-LCL in the summer afternoon over the TP is greater than that in eastern China. Compared to the eastern China, with the same relative humidity, lower temperature over the TP results in a lower lifting condensation level. With the same surface sensible heat flux, lower air density over the TP results in a larger buoyancy flux and a deeper boundary layer. The observational results show that, under low relative humidity condition (RH < 40%), the low cloud cover (LCC) is higher than 60% over the TP. In contrast, the high LCC (LCC > 60%) only appears under high RH condition (RH > 60%) at low elevation.

In general, LCC increases with increasing elevation. The median of LCCs at high elevation (TP) is significantly greater than those at low elevation (eastern China) throughout the day. The diurnal variations of LCC at low elevation are generally distributed in an unimodal pattern with the maximum appearing at 2:00 pm Beijing time and low values during the night. The diurnal variations of LCC at high elevation (TP) present a bimodal curve with the maximum appearing at 5:00 pm Beijing time and the secondary local maximum appearing at 8:00 am Beijing time. In addition, LCC maintains at high values at high elevation (TP) during the day. The median cloud top height derived from Himawari-8 retrieval product shows the transition from shallow clouds to deep convective clouds in the late afternoon and evening over the TP, which is attributed to the strong large-scale ascending motion from the near surface to upper troposphere over the TP.

The buoyancy term (BT) and shear term (ST) over the TP are significantly greater than those at the low elevation, which is favorable for the increasing of PBLH. Similar phenomena occur at other high elevation areas (e.g. Rocky Mountains). The strong thermal turbulence and large scale ascending motion jointly results in positive value of PBLH-LCL under low RH condition over the TP. The obvious large-scale subsidence on both sides of the Rocky Mountain especially the western side leads to inversion above PBL and lower RH within the PBL, which further lead to negative value of PBLH-LCL and decreased LCC in most part of Rocky Mountain. The slightly greater than zero PBLH-LCL corresponds spatially to increased LCC in the partial regions of central Rocky Mountain. Thus less LCC is generated at the Rocky Mountains compared to the TP.

## Data availability

All reanalysis data used in this study were obtained from publicly available sources: ERA5 reanalysis data can be obtained from the ECMWF public datasets web interface (http://apps.ecmwf.int/datasets/). The satellite (CloudSat radar and Calipso lidar)-merged cloud classification product 2B-CLDCLASS-lidar were obtained from

Colorado                          State                          University
(http://www.cloudsat.cira.colostate.edu/data-products/level-2b/2b-cldclass-lidar). The
Himawari-8 retrieval products were obtained from JAXA Himawari Monitor
(https://www.eorc.jaxa.jp/ptree/).

**Code Availability**

The data in this study are analysed with MATLAB and NCL. Contact Y.W. for specific

code requests.

**Acknowledgements**

Xu and Wang are supported by the Second Tibetan Plateau Scientific Expedition and
Research (STEP) program (Grant Nos. 2019QZKK0105), National Natural Science
Foundation of China (Grant Nos. 91837310), and the National Natural Science
Foundation for Young Scientists of China (Grant Nos. 41805006).

**Author Contributions**

X.X. and Y. W. led this work with contributions from all authors. Y.T. and Y. W.
made the calculations and created the figures. X.X., Y.W., H.Z. and M.Z. led analyses,
interpreted results and wrote the paper. R.L. supports high resolution satellite Gaofen
images to show the organized structures (cellular convection) for shallow convection.

**Competing interests**

The authors declare no competing interests.

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
