# Peer review of "Triggering effects of large topography and boundary layer turbulence on"

_Atmospheric Chemistry and Physics, 2022_

## Author Comment (AC1)

**Point-by-point responses to reviewers' comments**

We thank both reviewers for their detailed and constructive comments and suggestions. Following these comments and suggestions, we have
- done additional computations and provided more statistics in the discussion in Figure 7;
- added a paragraph in Section 2 to better descript the buoyancy term (BT) and shear term (ST) calculation method;
- added two panels to Figure S1;
- added a satellite image to show organized structures (cellular convection) in Figure 1.

Our revisions are indicated in the revised version with tracked changes. Below are our point-by-point responses (in blue).

Comment on acp-2022-221
Anonymous Referee #1
Comments on "Triggering effects of large topography and boundary layer turbulence over the Tibetan Plateau on convection"

General Comments:
The manuscript tries to analyze the diurnal variations and formation mechanism of low clouds at different elevations based on ERA5, the satellite cloud classification products and data sets from automatic weather stations from June to August of 2010-2019 in China. The author further discuss whether there exist triggering mechanism for convection over the Tibetan Plateau (TP), and whether there is an association among low air density, strong turbulence and ubiquitous "popcorn-like" cumulus clouds. The authors select two typical large topography regions (TP and Rocky mountains) to analyze the triggering effects of large topography and related dynamical structure within the boundary layer on convective clouds. Some interesting results have been obtained. I suggest that this manuscript could be accepted after minor revision.

Specific Comments and suggestion:
The writing of this manuscript needs to be improved. Such as the title maybe should change to "Triggering effects of large topography and boundary layer turbulence on convection over the Tibetan Plateau "
Done.

L110-113: The author used 0.25° x 0.25° ERA5 reanalysis data to calculate the buoyancy term (BT) and shear term (ST) in the TKE equation for each grid. How to interpret this method compared with traditional calculation of BT and ST in a micro-scale micrometeorology especially on the large TP terrain with strong heterogeneity? On the other hand, is it reasonable to used M-O similarity theory in

this grid?

Thank you for your comments. Of course, there exists uncertainty for the ERA5 reanalysis flux data especially on the large TP terrain with strong heterogeneity. However, in situ eddy covariance observations are too sparse to meet the needs of this study. Thus we use ERA5 reanalysis data to calculate the BT and ST. It is only an approximate calculation method of surface flux by using M-O similarity theory. Recent study (Xin et al., 2022, https://rmets.onlinelibrary.wiley.com/doi/epdf/10.1002/joc.7589) showed the bias errors of fluxes are generally smaller in ERA5 than in ERA-Interim. The Root mean square error between ERA5 flux data and in situ eddy covariance observations at eight sites over the TP are ranged from 21.82 W m$^{-2}$ to 46.73 W m$^{-2}$. We think this accuracy can meet the needs of this study. More studies need to evaluate the uncertainties of ERA5 flux data over the TP in future.

L150: "Figure 2 (a)" should change to Figure 2. In fig 2 caption "The monthly mean" should be "The summer mean"(June to Aug ).

Done.

Line 241 and 259: which is the relationship of BT and ST between calculated form the point measurement (such as soda) and from the ERA5 gird?

Compared to the 0.25° x 0.25° ERA5 reanalysis data, we think point measurement (such as soda) can reveal more local micro-scale information especially for the heterogeneous land surface.

Move the text in line 290-294 to line 203, and add more descriptions to show why the author select TP and Rocky Mountain as two typical large topography regions in subsequent paragraphs.

Thanks for your suggestion. We add more descriptions to illustrate this issue in line 228-234.

Line 297: Please show the range of latitude and longitude for TP and Rocky Mountain. Same for line 303.

Done.

---

## Author Comment (AC2)

**Point-by-point responses to reviewers' comments**

We thank both reviewers for their detailed and constructive comments and suggestions. Following these comments and suggestions, we have

- done additional computations and provided more statistics in the discussion in Figure 7;
- added a paragraph in Section 2 to better descript the buoyancy term (BT) and shear term (ST) calculation method;
- added two panels to Figure S1;
- added a satellite image to show organized structures (cellular convection) in Figure 1.

Our revisions are indicated in the revised version with tracked changes. Below are our point-by-point responses (in blue).

Comment on acp-2022-221 Anonymous Referee #2

**General**

This paper analyses cumulus cloud cover over China with a focus on the difference between the Tibetan Plateau and regions with less topography. Finally, results are compared with the North American region. It is found that topography has a triggering effect which is more pronounced over the Tibetan Plateau than over the Rocky Mountains because of the larger impact of subsidence in the latter region. This is in principle an interesting topic, but I find that the presentation needs much improvement before its publication. My major concerns and some minor points are described below.

**Major revisions**

The considered topic is not new and the differences to existing literature should be better described. New findings should become clearer. Especially, the differences to Wang et al. (2020) need to be explained who also studied the Tibetan cloud cover. Figure 6 is shown in the same way in Wang et al. (2020) but this is not mentioned. What is new here?

Figure 6 (a) is basically similar to the Figure 1 (a) in Wang et al. (2020). We add a paragraph in line 306-313 to show more new findings in Figure (b)-(d).

I have difficulties to understand the principle idea. Why should the TKE budget at the surface play the most important role for cloud cover? I can follow that the near-surface buoyancy flux is important and also the near-surface shear stress is important for the PBL height, but there are many other impact factors influencing clouds such as aerosol, large scale forcing etc. Also, there are other sources of turbulence especially at cloud top and condensation level which might have an impact.

The main purpose of introducing TKE budget equation is to show the specific forms of buoyancy and shear terms (BT and ST), and then we use ERA5 reanalysis data to calculate BT and ST. Here we do not think all the terms in TKE budget equation play an important role for cloud cover. We agree with your comments that other factors (e.g. aerosol, large scale forcing) also play a key role in clouds formation and development. As shown in Figure 3, compared to the Rocky Mountains, the obvious large scale ascending motions over the TP are in favour of clouds formation and development. We also discussed the variations of PBLH-LCL on clouds. Please refer to the relevant paragraph for more details. Other factors such as aerosol are not mentioned in this study, further data analysis is needed to elucidate the role of these factors.

Before equation (3) occurs, it must be clearly said that in the following the determination (iterative scheme) of the surface fluxes is explained. But the equations are incomplete. The characteristic temperature scale (theta\_star occurring in the Obukhov length) must be involved, otherwise the system cannot be solved and neither friction velocity nor heat flux can be determined. I guess, equation (6) is for heat? Equation (7) does not involve humidity, which is in contrast to equation (3).

Thank much for this comment. Yes. The heat flux is derived from  $\theta_*$  by using M-O similarity theory. Here we directly use ERA5 reanalysis sensible heat flux product, and then use equation (3) to derive  $\overline{w'\theta'_v}$ . The equation (6) is for momentum rather than heat, here we do not show the  $\phi_h$  for heat. We add a subscript v for  $\theta'$  in equation (7).

It is several times repeated that there are organized structures (cellular convection) (e.g. in lines 162, 163, 231). What is the basis for this conclusion? I expected at least a satellite image showing the typical cell structure and the cumulus clouds which are described as 'popcorn-like'.

Thank you for your suggestion. We add a co-author who supports high resolution satellite Gaofen 4 images to show the organized structures (cellular convection) for shallow convection.

When the goal is to compare results in China with those in North America then a similar Figure 2 should be shown for North America.

Figure 2 are derived from in situ measurements LCC in China, we do not show a similar figure in North America due to lack of this kind of data in North America. For comparing, we also plot Figure 7 (e) and (f) to show the summer mean LCC derived from cloudsat satellite data at local time 2:00 pm in Eastern Asia and North America.

Please explain results showing wind vectors in Figure 4. There is no unit given, but at present I must conclude that mean vertical velocities are in the order of 4 m/s (at least the same order as horizontal wind). But they should be close to zero. Or what is the reason for the permanent strong upward wind over the Tibetan Plateau?

The length of wind vectors in Figure 4 cannot denote the actual wind speed due to the different orders of magnitude for the horizontal or vertical velocities, thus we delete the legend in Figure 4. In order to highlight the large scale ascending or descending region in Figure 4, we extend the vertical velocities by 100 times. Figure 4 show the

summer mean large scale vertical velocities, TP as a heat source in summer, there is strong upward wind over the TP, which correspond the convergence in middle troposphere (about 500 hPa) and the divergence in upper troposphere (about 200 hPa). The definition of the PBL is unclear. In Figure 4, it seems that over long distances

LCL and PBL are at the same level. But usually, shallow cumulus at least is part of the PBL.Cloud base is at LCL but the rest of the cloud above it.

Here we directly use the PBLH product from ERA5 reanalysis data. We agree your comments. Sorry for the unclear figure 3 captions. Figure 4 only show the summer mean PBL height and LCL at local time 2:00 pm, thus over long distances (e.g. eastern China) LCL and PBLH are almost at the same level.

Figure 8: According to the figure, the authors seem to consider deep convection. But this is not clear from the beginning of the paper. 'Cumulus convection' is referring to shallow convection as well. Please specify already in the introduction, which kind of convection is considered. Figure 8 would give a wrong impression when the paper addresses also shallow convection.

Yes. We discuss and analyze both the shallow and deep convention in this study. Figure 5 show the spatial distribution of day time variations of cloud top height in summer, which reflect the evolution from shallow convection to deep convection over the TP. Compared to the eastern China, higher median cloud top height in summer implies that deep convection are more likely to occur over the TP.

Mínor revisions

Line 54: why does decreasing RH favors the formation of clouds? Sorry for the mistakes. It should be "increasing".

Line 96: replace 'obscured' by 'covered'

Done.

Line 150: add that the figure is based on reanalysis

Done.

Line 152: what is an 'in ribbon' pattern?

For the purpose of expressing more clearly, we delete the words "ribbon pattern", and revise the sentence.

Line 152: better show a map with the Tibetan Plateau and Yangtze River valley or add this explanation in an existing figure

Thanks for your suggestion. We add a sentence in Figure 2 caption.

Figure 3: Explain all abbreviations (ASL, AGL and others). Show this figure also for North America.

We have revised all abbreviations, and added necessary explanation. Using ERA5 LCC data, we add Figure S1 to show the diurnal cycle of LCC in summer in East Asia and North America in supplementary material. It should be noted that there exists some differences between the LCC from ERA5 and in situ measurements due to the different definition and model deviation. ERA5 defines low clouds as those between surface and the height at 80% of the surface pressure (or the lowest ~2 km).

---

## Referee Report (RR1)

General

The paper was improved, and the authors gave answers to my questions, but several points need further clarification before I can recommend the paper's publication. There are still many language issues, and I think this version requires still major English Editing.

In the following, I refer to line numbers of the version with marked changes.

**Major revisions**

1) The paper by Wang et al. (2020)  has a very similar topic. It would strengthen the paper when in the introduction the differences of goals to those of the new paper would become clearer. I guess, the main difference is the comparison with North America, but perhaps there are others?

2) The hypothesis on the role of the TKE budget terms is better explained in Wang et al (2020). So, please refer also here to the Wang et al. (2020) paper. It can serve as a motivation for your study in the introduction!

3) Description of the determination of wind shear (equations 4-10): I strongly recommend writing L directly as a function of $\tau$  and H where H is heat flux (see equation 1 in Gryanik et al. 2020). Then one can tell that after prescribing values for $\tau$ and H from the model,  L is determined and then via equation 4 the shear term. However, this method contains an inconsistency. Namely, when ERA5 takes other similarity functions than those of Dyer (1974). Please check this. If yes, then the shear does not correspond to the model equations and is just an approximation. This drawback needs to be explained.

4) After adding the 2500m contour lines in Figure 2, I see that not the whole TP region has high LCC, but roughly one third of the region is not concerned. This should be discussed!

5) I asked to explain results showing wind vectors (now Figure 4a and 4b). But I am not satisfied with the answer that the legend is now simply skipped. So, are the wind vectors now only a schematic?  One needs to understand the effect quantitatively. Please note that this is one of the most important results (the central figure) for explaining the different LCC in the North American and Asian region. This needs explanation in the paper, not just for me!

6) I had asked for the discrepancy concerning the PBL and LCL (now in Figure 4). The authors answered this but this caused no change in the text (or did I oversee this?) Without further explanation, I would conclude from Figure 4 c) and d) that something is wrong with the definition of PBL since LCL cannot be so close to PBL over huge distances. This needs explanation in the text.

7) Language: in principle, the text can be understood but there are still many errors in almost every third sentence (smaller ones with just wrong articles but also larger ones with grammar and wording).

**Mínor revisions**

Line 48: I think acronym TIPEX was not yet explained here.

Line 49: In which paper, the term popcorn like cumulus clouds was used the first time for this area? Please cite.

Line 69: sentence starting with 'according to'. It is too long and could be better understood by splitting the information into two sentences.

Figure 1 Caption needs to be changed. Is cloud fraction shown? Just 'digital number' is not enough information.

Figure 2: delete second occurrence of summer in the caption. Blue line is hard to see, replace it by white?1) Which cloud characteristic is changing? What is 'raised' cloud?

Line 174: I cannot understand the sentence. What is shown in Figure

Lines 178-180: Indices L, M,H are not explained.

Figure 7: write e) and f) in the same size as a) b) c) d).

Line 307-313. These sentences need language revision. It is difficult to follow. But I am also afraid that the difference PBLH-LCL should not be interpreted at al all (see major point 6).

Reference

Gryanik, V. M., Lüpkes, C., Grachev, A., & Sidorenko, D. (2020). New modified and extended stability functions for the stable boundary layer based on SHEBA and parametrizations of bulk transfer coefficients for climate models. *Journal of the Atmospheric Sciences*, *77*(8), 2687-2716.

---

## Referee Report (RR2)

The paper was improved once more, and the authors answered my questions and did corresponding modifications. But as the text improved I was more able to follow and found now a few more things requiring revision. Nevertheless, although these are also major points, I guess that after improvement the paper might be in a form that can be published.

In the following, I again refer to line numbers of the first revised version with marked changes.

**Major Revisions**

1. In my review of the revised version I asked the following question: The paper by Wang et al. (2020) has a very similar topic. It would strengthen the paper when in the introduction the differences of goals to those of the new paper would become clearer. I guess, the main difference is the comparison with North America, but perhaps there are others?

   The authors answered that question well but as far as I can see this did not cause any modification in their manuscript. The answer should occur in the introduction. Only then, the reader is able to understand the novelty of the study at the beginning.

2. I am still not satisfied with the description of the method between lines 149 and 177. The present form cannot be understood. Once more, I strongly recommend the following:

   Write that wind shear is determined from heat flux $H$ and momentum flux $\tau$ obtained from the ERA5 reanalysis data. Namely, according to Monin Obukhov similarity theory wind shear is given as

   $$\frac{\partial \overline{u}}{\partial z} = \phi_m(\zeta)\frac{u_\star}{\kappa z} \tag{1}$$

   where $\phi_m$ is the Monin Obukhov stability function for momentum, $u_\star^2 = \tau/\rho$. $\zeta = z/L$ with $z =$ height and $L =$ Obukhov stability length defined as in Gryanik et al. (2020) as

   $$L = -\frac{(\tau/\rho)^{3/2}}{\kappa(g/\theta_v)(H/\rho c_p)} \ . \tag{2}$$

   $\phi_m$ is the Monin Obukhov stability function where we used

   $\phi_m = ....$   your old equations (5) and (6) for stable and unstable conditions $\quad$ (3)

No further equations are necessary. When you followed the above procedure this must be explained in this way, if something else was done it would need a better description.

3. Still, the quality of some figures is bad. These are Figure 4 (labels of colour bars in a) and b) almost not readable, labels are not readable of 4 c and d). Figure 6 (text in black boxes very difficult to read, increase resolution). Figure 7a, e,f: all labels should have the same size as in 7b.

**Minor revisions**

Abstract: the text of the conclusions is much better than the text of the abstract. The minimum modifications are: line 24: correct to 'with increasing difference PBLH-LCL'

Lines 24-28: I suggest writing: The triggering of convection by boundary layer dynamics is analyzed over TP but also in the Northern Hemisphere over the Rocky Mountains. It is found that ST and BT are strong over both high elevation regions ...

line 32: write... by inversions above the PBL and to lower RH within the PBL, which further leads
line 34: at the Rocky Mountains
Line 44: It is a dynamic effect caused by the
line 87: of a cumulus system
line 88: in the PBL
line 89: with anomalous
line 90: processes
line 118: with a spatial
line 127: averaged
line 199: with increasing
line 232: median
line 240: dashed contour
line 250: from the himawari
line 255: trend of decreasing LCC
lines 244-245: Verb is missing in sentence
Lines 262-264: I do not understand why 200 hPa is compared with 500 hPa. This needs more explanation. Describe exactly where you see divergence, where convergence.
Line 270: to the middle
Line 272: the inversion is not really seen in the figure
Line 273: to an increased
Line 279: one needs a reference with respect to CISK
Line 281: the Western TP

Line 283: in the northern

Line 285: what's a foke low?

Line 285: in the northern

Line 299: areas

Line 306: of the convective

Line 309: reformulate sentence, that it becomes clearer that BT and ST play a key role (and not the elevation)

line 317: for the North Sea (or over North Sea)

line 324; 2015). Thus one might ask the question what is ...

line 333; which is consistent with

line 355: low elevation regions.. start new sentence after afternoons

line 366: to two mechanisms. Now start with The first mechanism ,,, and later the second mechanism ....

line 364: The blue and red histograms show the surface elevation (blue) and temperature (red) as functions of 2 m air density

line 373: shows

line 375: values

line 383: which refelect special surface characteristics in the boundary

line 387: shows a conceptual ...of the atmosphere

line 413: TP plays a

line 415: found that the difference PBLH-LCL

line 426: in an unimodal

line 437: phenomena

line 483: the name is De Bruin, not just Bruin (see also citation in the text)

line 423: with increasing

---

## Editor Decision (ED1)

The paper was improved once more, and the authors answered my questions and did corresponding modifications. But as the text improved I was more able to follow and found now a few more things requiring revision. Nevertheless, although these are also major points, I guess that after improvement the paper might be in a form that can be published.

In the following, I again refer to line numbers of the first revised version with marked changes.

**Major Revisions**

1. In my review of the revised version I asked the following question: The paper by Wang et al. (2020) has a very similar topic. It would strengthen the paper when in the introduction the differences of goals to those of the new paper would become clearer. I guess, the main difference is the comparison with North America, but perhaps there are others?

   The authors answered that question well but as far as I can see this did not cause any modification in their manuscript. The answer should occur in the introduction. Only then, the reader is able to understand the novelty of the study at the beginning.

2. I am still not satisfied with the description of the method between lines 149 and 177. The present form cannot be understood. Once more, I strongly recommend the following:

   Write that wind shear is determined from heat flux $H$ and momentum flux $\tau$ obtained from the ERA5 reanalysis data. Namely, according to Monin Obukhov similarity theory wind shear is given as

   $$\frac{\partial \overline{u}}{\partial z} = \phi_m(\zeta)\frac{u_\star}{\kappa z} \tag{1}$$

   where $\phi_m$ is the Monin Obukhov stability function for momentum, $u_\star^2 = \tau/\rho$. $\zeta = z/L$ with $z =$ height and $L =$ Obukhov stability length defined as in Gryanik et al. (2020) as

   $$L = -\frac{(\tau/\rho)^{3/2}}{\kappa(g/\theta_v)(H/\rho c_p)} \ . \tag{2}$$

   $\phi_m$ is the Monin Obukhov stability function where we used

   $$\phi_m = .... \quad \text{your old equations (5) and (6) for stable and unstable conditions} \tag{3}$$

No further equations are necessary. When you followed the above procedure this must be explained in this way, if something else was done it would need a better description.

3. Still, the quality of some figures is bad. These are Figure 4 (labels of colour bars in a) and b) almost not readable, labels are not readable of 4 c and d). Figure 6 (text in black boxes very difficult to read, increase resolution). Figure 7a, e,f: all labels should have the same size as in 7b.

**Minor revisions**

Abstract: the text of the conclusions is much better than the text of the abstract. The minimum modifications are: line 24: correct to 'with increasing difference PBLH-LCL'

Lines 24-28: I suggest writing: The triggering of convection by boundary layer dynamics is analyzed over TP but also in the Northern Hemisphere over the Rocky Mountains. It is found that ST and BT are strong over both high elevation regions ...

line 32: write... by inversions above the PBL and to lower RH within the PBL, which further leads
line 34: at the Rocky Mountains
Line 44: It is a dynamic effect caused by the
line 87: of a cumulus system
line 88: in the PBL
line 89: with anomalous
line 90: processes
line 118: with a spatial
line 127: averaged
line 199: with increasing
line 232: median
line 240: dashed contour
line 250: from the himawari
line 255: trend of decreasing LCC
lines 244-245: Verb is missing in sentence
Lines 262-264: I do not understand why 200 hPa is compared with 500 hPa. This needs more explanation. Describe exactly where you see divergence, where convergence.
Line 270: to the middle
Line 272: the inversion is not really seen in the figure
Line 273: to an increased
Line 279: one needs a reference with respect to CISK
Line 281: the Western TP

Line 283: in the northern

Line 285: what's a foke low?

Line 285: in the northern

Line 299: areas

Line 306: of the convective

Line 309: reformulate sentence, that it becomes clearer that BT and ST play a key role (and not the elevation)

line 317: for the North Sea (or over North Sea)

line 324; 2015). Thus one might ask the question what is ...

line 333; which is consistent with

line 355: low elevation regions.. start new sentence after afternoons

line 366: to two mechanisms. Now start with The first mechanism ,,, and later the second mechanism ....

line 364: The blue and red histograms show the surface elevation (blue) and temperature (red) as functions of 2 m air density

line 373: shows

line 375: values

line 383: which refelect special surface characteristics in the boundary

line 387: shows a conceptual ...of the atmosphere

line 413: TP plays a

line 415: found that the difference PBLH-LCL

line 426: in an unimodal

line 437: phenomena

line 483: the name is De Bruin, not just Bruin (see also citation in the text)

line 423: with increasing

Referee #2

Most of my concerns have been addressed, and the authors revised the manuscript according the comments and suggestions proposed by referees properly. I do not hesitate to suggest accepting it for publication as long as the following minor points are considered.

Issues:

1. L29: Please introduce "RH" when it first appears.

2. L121: himawari-8 -> Himawari-8. Please check the entire manuscript.

3. L132 and L136: The definitions for shear term are different. Please check which equation is actually used in this work.

4. L207: "smaller" or "larger"? How did the authors make this argument? Based on which plot?

5. Figure 4: Please indicate the units of divergence in the colorbars in Figure 4a and 4b. The resolution of the figures needs to be improved, as it is difficult to see the small numbers.

6. L254-256: I suggest deleting "compared to eastern China", as one do not see the decrease trend in LCC in eastern China. Otherwise, the authors should provide a reference or a figure to show the decrease in LCC from late afternoon to evening in eastern China.

7. L294: "the second Tibet Plateau Experiment (TIPEX II)" -> "the TIPEX II", and check the entire manuscript for the same issue.

8. Figure 7: You have two (e)s in the caption. One sub-plot is missing (the first figure 7e)? Please add colorbar for figure 7a, and add units for all color scales.

9. L370: "two typical high value regions……". High positive value, negative value, or absolute value?

10. L425: Use "at low elevations" other than "in eastern China" is more accurate.

11. L438-: The positive value of PBLH-LCL is only over the TP, not over the Rocky Mountains. And the PBLH-LCL is -101.9 m, not slightly greater than zero. Please revise your conclusions.

---

## Author Response (AR2)

**Point-by-point responses to reviewer2's comments**

We thank reviewer 2 for his (or her) detailed and constructive comments and suggestions. Following these comments and suggestions, we have
- added a satellite image (Figure 1b) to show the spatial distribution of cloud in southeastern TP.
- revise the paragraph in Section 2 to better descript the shear term (ST) calculation method, and re-calculate shear term;
- added Figure S2 to show the spatial distribution of daily mean vertical velocities W at 500 hPa in summer over land in East Asia;
- done additional computations and provided more statistics about PBLH and LCL in the discussion in Figure 4c and 4d;

Our revisions are indicated in the revised version with tracked changes. Below are our point-by-point responses (in blue).

General

The paper was improved, and the authors gave answers to my questions, but several points need further clarification before I can recommend the paper's publication. There are still many language issues, and I think this version requires still major English Editing.

In the following, I refer to line numbers of the version with marked changes.

**Major revisions**

1) The paper by Wang et al. (2020) has a very similar topic. It would strengthen the paper when in the introduction the differences of goals to those of the new paper would become clearer. I guess, the main difference is the comparison with North America, but perhaps there are others?

Yes. The main difference is the comparison with North America. In addition, we focus on the effect of large scale vertical velocities on clouds. There are significant large scale ascending motions in most parts of TP (except part of western TP and Qaidam Basin with descending motions) as shown in Figure S2, which lead to less LCC in northwestern TP and Qaidam Basin.

2) The hypothesis on the role of the TKE budget terms is better explained in Wang et al (2020). So, please refer also here to the Wang et al. (2020) paper. It can serve as a motivation for your study in the introduction!

Thank you for your suggestion. We add the two key points proposed by Wang et al. (2020) in the introduction.

3) Description of the determination of wind shear (equations 4-10): I strongly recommend writing L directly as a function of $\tau$ and H where H is heat flux (see equation 1 in Gryanik et al. 2020). Then one can tell that after prescribing values for $\tau$ and H from the model, L is determined and then via equation 4 the shear term. However, this method contains an inconsistency. Namely, when ERA5 takes other similarity functions than those of Dyer (1974). Please check this. If yes, then the shear does not correspond to the model equations and is just an approximation. This drawback needs to be explained.

Thank you for your suggestion. We have revised the description of the determination
of wind shear. Based on the ERA5 physical processes documentation, we re-calculate the
shear term by using the similarity functions under unstable (or stable) condition. We think
the new shear term results will be more consistent with ERA5 model.

4) After adding the 2500m contour lines in Figure 2, I see that not the whole TP
region has high LCC, but roughly one third of the region is not concerned. This should be
discussed!

Thank you for your comments. We add the spatial distribution of the vertical
velocities at 500 hPa in Figure S2. We also add some discussions about the effect of
vertical velocities on clouds over the TP, and explain why there are high LCC in northern
part of TP (80-90E, 34-36N) in Figure 2 in line 272-278.

5) I asked to explain results showing wind vectors (now Figure 4a and 4b). But I am
not satisfied with the answer that the legend is now simply skipped. So, are the wind
vectors now only a schematic? One needs to understand the effect quantitatively. Please
note that this is one of the most important results (the central figure) for explaining the
different LCC in the North American and Asian region. This needs explanation in the
paper, not just for me!

The vertical velocities are not of the same magnitude as the horizontal velocities,
so wind vectors are only a schematic. In order to show the true values, we remove the
vectors in Figure 4a and 4b. Now we use black and green contours to show the W-
and U- wind components, respectively.

6) I had asked for the discrepancy concerning the PBL and LCL (now in Figure 4).
The authors answered this but this caused no change in the text (or did I oversee this?)
Without further explanation, I would conclude from Figure 4 c) and d) that something is
wrong with the definition of PBL since LCL cannot be so close to PBL over huge
distances. This needs explanation in the text.

The original Figure 4 (c) and (d) only show the mean PBLH and LCL at 2:00 pm
local time, thus the deviations between mean PBLH and LCL are generally less than 500
m (or 50 hPa). The y-axis ranges from 1000 hPa to 300 hPa, thus LCL seems to be close
to the PBLH. To avoid the above misunderstanding, we plot the new Figure 4 (c) and (d)
with the purpose of showing the PBLH and LCL versus longitude in East Asia and North
America, respectively.

7) Language: in principle, the text can be understood but there are still many errors
in almost every third sentence (smaller ones with just wrong articles but also larger ones
with grammar and wording).

Done. We invite a cooperator to revise the paper.

**Mínor revisions**

Line 48: I think acronym TIPEX was not yet explained here.

Done.

Line 49: In which paper, the term popcorn like cumulus clouds was used the first
time for this area? Please cite.

Done.

Line 69: sentence starting with 'according to'. It is too long and could be better understood by splitting the information into two sentences.

Done.

Figure 1 Caption needs to be changed. Is cloud fraction shown? Just 'digital number'

is not enough information.

We have revised the Figure 1 caption, and add one panel (Figure 1b) to show the spatial distribution of cloud. We also calculate the mean cloud fraction over the southeastern part of TP (92.7-96.2E, 29.5-31.3N).

Figure 2: delete second occurrence of summer in the caption. Blue line is hard to see, replace it by white? 1) Which cloud characteristic is changing?

Done.

Line 174: I cannot understand the sentence. What is shown in Figure? What is

'raised' cloud?

We revise the sentence, and delete the unclear expression "raised".

Lines 178-180: Indices L, M, H are not explained.

Thanks. We add the explanation in Figure 3 caption.

Figure 7: write e) and f) in the same size as a) b) c) d).

Done.

Line 307-313. These sentences need language revision. It is difficult to follow. But I

am also afraid that the difference PBLH-LCL should not be interpreted at al all (see major point 6).

Done. We have revised the sentences, and answered point 6.

---

## Author Response (AR3)

Point-by-point responses to two reviewers' comments

We thank two reviewers for their detailed and constructive comments and suggestions. Following these comments and suggestions, we have
- revise the paragraph in Section 2 to better descript the shear term (ST) calculation method;
- revise the abstract and conclusions to better illustrate our ideas;
- added Figure S2 to show the inversion above the PBL and lower RH in near surface layer over the Rocky Mountains;
- revise the unclear Figures (e.g. Figure 4 and 7);

Our revisions are indicated in the revised version with tracked changes. Below are our point-by-point responses (in blue).

Referee #2

General

The paper was improved once more, and the authors answered my questions and did corresponding modifications. But as the text improved I was more able to follow and found now a few more things requiring revision. Nevertheless, although these are also major points, I guess that after improvement the paper might be in a form that can be published. In the following, I again refer to line numbers of the first revised version with marked changes.

Major Revisions

1. In my review of the revised version I asked the following question: The paper by Wang et al. (2020) has a very similar topic. It would strengthen the paper when in the introduction the differences of goals to those of the new paper would become clearer. I guess, the main difference is the comparison with North America, but perhaps there are others?
The authors answered that question well but as far as I can see this did not cause any modification in their manuscript. The answer should occur in the introduction. Only then, the reader is able to understand the novelty of the study at the beginning.

Thank you for your suggestions. We add this key point in introduction.

2. I am still not satisfied with the description of the method between lines 149 and 177. The present form cannot be understood. Once more, I strongly recommend the following:

Write that wind shear is determined from heat flux H and momentum flux $\tau$

obtained from the ERA5 reanalysis data. Namely, according to Monin Obukhov similarity theory wind shear is given as

$$\frac{\partial \bar{u}}{\partial z} = \phi_m(\zeta)\frac{u_*}{kz},$$

(1)

where $\phi_m$ is the Monin Obukhov stability function for momentum, $u_*^2 = \tau/\rho$. $\zeta = z/L$ with $z$ = height and $L$ = Obukhov stability length defined as in Gryanik et al. (2020) as

$$\zeta = \frac{z}{L}, L = -\frac{(\tau/\rho)^{3/2}}{\kappa(g/\theta_v)(H/\rho c_p)}.$$

(2)

$\phi_m$ is the Monin Obukhov stability function where we used $\phi_m = $ …. your old equations (5) and (6) for stable and unstable conditions (3)

No further equations are necessary. When you followed the above procedure this must be explained in this way, if something else was done it would need a better description.

Thank you for your suggestion. We revised the description of the determination of wind shear again.

3. Still, the quality of some figures is bad. These are Figure 4 (labels of colour bars in a) and b) almost not readable, labels are not readable of 4 c and d). Figure 6 (text in black boxes very difficult to read, increase resolution). Figure 7a, e,f: all labels should have the same size as in 7b.

Thanks. We have revised all the unclear figures.

Minor revisions

Abstract: the text of the conclusions is much better than the text of the abstract.

We have revised the abstract again.

The minimum modifications are:

Line 24: correct to 'with increasing difference PBLH-LCL'

Sorry for our mistake. We delete this incorrect sentence.

Lines 24-28: I suggest writing: The triggering of convection by boundary layer dynamics is analyzed over TP but also in the Northern Hemisphere over the Rocky Mountains. It is found that ST and BT are strong over both high elevation regions.

Done.

Line 32: write... by inversions above the PBL and to lower RH within the PBL, which further leads

Done.

Line 34: at the Rocky Mountains

Done.

Line 44: It is a dynamic effect caused by the

Done.

line 87: of a cumulus system

Done.

line 88: in the PBL

Done.

line 89: with anomalous

Done.

line 90: processes

Done.

line 118: with a spatial

Done.

line 127: averaged

Done.

line 199: with increasing

Done.

line 232: median

Thanks. But we think "medium" is more suitable word.

line 240: dashed contour

Done.

line 250: from the himawari

Done.

line 255: trend of decreasing LCC

Done.

lines 244-245: Verb is missing in sentence

Thanks. We add it in sentence.

Lines 262-264: I do not understand why 200 hPa is compared with 500 hPa. This needs more explanation. Describe exactly where you see divergence, where convergence.

The average altitude of the TP is 4000 m (~600 hPa). The 500 hPa corresponds to the lower atmosphere layer (or middle troposphere) over the TP, and the 200 hPa roughly corresponds to the upper troposphere. The convergence in the middle troposphere and the divergence in the upper troposphere are usually associated with the deep convection over the TP.

Line 270: to the middle

Done.

Line 272: the inversion is not really seen in the figure

We add a Figure S2 in supplementary material to show the inversion above the PBL and lower RH within the PBL at both sides of Rocky Mountains.

Line 273: to an increased

Done.

Line 279: one needs a reference with respect to CISK

Thanks. We have added it.

Line 281: the Western TP

Done.

Line 283: in the northern

Done.

Line 285: what's a fake low?

We replace fake with false.

Line 285: in the northern

Done.

Line 299: areas

Done.

Line 306: of the convective

Done.

Line 309: reformulate sentence, that it becomes clearer that BT and ST play a key role (and not the elevation)

Done.

line 317: for the North Sea (or over North Sea)

Done.

line 324; 2015). Thus one might ask the question what is ...

Done.

line 333; which is consistent with

Done.

line 335: low elevation regions.. start new sentence after afternoons

Done.

line 366: to two mechanisms. Now start with The first mechanism ,,, and later the second mechanism ....

Done.

line 364: The blue and red histograms show the surface elevation (blue) and temperature (red) as functions of 2 m air density

Done.

line 373: shows

Done.

line 375: values

Done.

line 383: which reflect special surface characteristics in the boundary

Done.

line 387: shows a conceptual ...of the atmosphere

Done.

line 413: TP plays a

Done.

line 415: found that the difference PBLH-LCL

Done.

line 426: in an unimodal

Done.

line 437: phenomena

Done.

line 483: the name is De Bruin, not just Bruin (see also citation in the text)

Thank you for your comments. We use the old equation for unstable condition, so we delete related content in the main text and References.

line 423: with increasing

Done.

Referee #3

General

Most of my concerns have been addressed, and the authors revised the manuscript according the comments and suggestions proposed by referees properly. I do not hesitate to suggest accepting it for publication as long as the following minor points are considered.

Issues:

1. L29: Please introduce "RH" when it first appears.

Done.

2. L121: himawari-8 -> Himawari-8. Please check the entire manuscript.

Done.

3. L132 and L136: The definitions for shear term are different. Please check which equation is actually used in this work.

Done.

4. L207: "smaller" or "larger"? How did the authors make this argument? Based on which plot?

Sorry for our mistake. It should be "larger " based on the Figure 3.

5. Figure 4: Please indicate the units of divergence in the colorbars in Figure 4a and

4b. The resolution of the figures needs to be improved, as it is difficult to see the small numbers.

Thanks for your suggestion. We have revised this figure.

6. L254-256: I suggest deleting "compared to eastern China", as one do not see the decrease trend in LCC in eastern China. Otherwise, the authors should provide a reference or a figure to show the decrease in LCC from late afternoon to evening in eastern China.

Thanks. We delete "compared to eastern China".

7. L294: "the second Tibet Plateau Experiment (TIPEX II)" -> "the TIPEX II", and check the entire manuscript for the same issue.

Done.

8. Figure 7: You have two (e)s in the caption. One sub-plot is missing (the first figure

7e)? Please add colorbar for figure 7a, and add units for all color scales.

Thanks. We have revised the Figure caption.

9. L370: "two typical high value regions……". High positive value, negative value, or absolute value?

Sorry for our mistake that we did not express our ideas clearly. We have replaced

"high value" with "large topography".

10. L425: Use "at low elevations" other than "in eastern China" is more accurate.

Done.

11. L438-: The positive value of PBLH-LCL is only over the TP, not over the Rocky

Mountains. And the PBLH-LCL is -101.9 m, not slightly greater than zero. Please revise your conclusions.

Thank you for your comments. We have revised the conclusions.